# Analyzing the Effect of Yellow Safety Line Designs at the Platform Edge in Metro Stations: An Experimental Approach

Sebastian Seriani [1,*], Pablo Arce [1], Carla Belmar [1], Vicente Blanche [2], Alejandra Valencia [1], David Luza [3] and Taku Fujiyama [4]

1. Escuela de Ingeniería de Construcción y Transporte, Pontificia Universidad Católica de Valparaíso, Valparaíso 2362804, Chile; pablo.arce.p@mail.pucv.cl (P.A.); carla.belmar.m@mail.pucv.cl (C.B.); alejandra.valencia@pucv.cl (A.V.)
2. Facultad de Ingeniería y Ciencias Aplicadas, Universidad de los Andes, Santiago 7620001, Chile; vblanche@miuandes.cl
3. Escuela de Arquitectura y Diseño, Pontificia Universidad Católica de Valparaíso, Valparaíso 2580129, Chile; david.luza@ead.cl
4. Faculty of Civil, Environmental and Geomatic Engineering, University College London, Gower St., London WC1E 6BT, UK; taku.fujiyama@ucl.ac.uk
* Correspondence: sebastian.seriani@pucv.cl

**Abstract:** The objective of this paper was to analyze the effect of yellow-safety-line designs on the behavior of passengers at the platform edge in metro stations. To achieve this, an experimental approach, based on observation, was used in existing metro stations in Santiago and Valparaiso, Chile. The experiments were carried out for different widths of the yellow safety line: 5 cm, 10 cm, 24 cm, and 40 cm. In addition, the material was also changed to include yellow adhesive tape, PVC material with yellow pods, and carbon- and fiberglass-reinforced material with yellow pods. The experiments considered a mock-up to represent the hall entrance of the train and its adjacent platform, in which 25 participants were recruited, some of whom had reduced mobility. The results obtained from the experiments showed that the greater the width of the yellow safety line at the edge of the platform, the greater the level of compliance that was achieved. In addition, surveys were carried out with the passengers who participated in the experiment; the majority felt more comfortable and safer for a width of 24 cm. Some participants highlighted the phenomenon of "safety offers comfort". In conclusion, the results of this research will allow the generation of new design and safety standards for the train–platform interface, which can then be tested in existing stations. Future research is expected to study the space occupied by different types of passengers and to study accessibility in other circulation spaces of metro stations.

**Keywords:** passenger; reduced mobility; accessibility; safety; yellow line; platform–train interface; laboratory experiments; metro station

## 1. Introduction

Metro stations can be studied for different pedestrian circulation spaces: train–platform, platform–stairs, mezzanines, complementary spaces (for example, commerce), and city (street-level). However, the space where most interactions take place is the train–platform interface, where passengers get on and off [1]. When the platform–train interface does not have an adequate design, passengers must travel long distances and move within unsafe spaces.

In the case of the metro in Valparaíso, more than 20 million interactions take place at the platform–train interface each year, reaching around 60 thousand passengers each day who board or alight the train [2]. In a more congested system, such as the Santiago Metro, there are around 2.8 million interactions daily between passengers boarding and alighting [3]. Similarly, in the case of other systems worldwide such as the National Network in the

United Kingdom, each year there are more than 3 billion interactions on the train network, and 48% of the passenger fatality risks occur at the platform–train interface [4]. Therefore, this complex space presents different risks and dangers for passengers. Accidents can happen while getting on and off or just on the edge of the platform when passengers are waiting for the train to arrive.

In the case of Santiago, a report prepared by Metro de Santiago showed that between 2017 and March 2019 there were 54 suicide attempts in stations, of which 20 were fatal and the rest were frustrated [5]. An even more worrying figure shows how, between 2017 and 2019, these cases increased by 39%, with 55% of them being passengers between 18 and 30 years old. These suicides represent a type of psychological disability.

To improve safety conditions at this interface, different metro systems have implemented accessibility measures at stations. One example is the platform doors, which prevent passengers from falling onto the train lines and make it possible to identify where each train door is located [6–8]. In the case of Santiago, the new Lines 3 and 6 have this type of door; they are 2.0 m wide and open at the same time as the train doors. When it is not possible to implement platform doors, other accessibility measures are required to promote access and use of public transport [9,10]. For example, the platform–train interface of Line 1 at the Santiago Metro has a yellow line at the edge of the train, to caution passengers from approaching the edge of the platform. Although these measures are used mainly for safety reasons, the effect they have on passenger behavior is unknown, and therefore there is a lack of design and safety standards. In particular, there is variability in the width and material of the yellow line, which indicates a lack of a safety standard for this space. In the case of Santiago, the metro system presents different variations in width: 5 cm, 10 cm, 20 cm, and 24 cm (see Figure 1). In these cases, the yellow safety line is complemented by other accessibility measures such as tactile pavement. Therefore, some yellow safety lines are only generated using a 5 cm or 10 cm yellow adhesive tape, while there are other designs (e.g., 20 cm and 24 cm widths) in which PVC material with yellow pods is used.

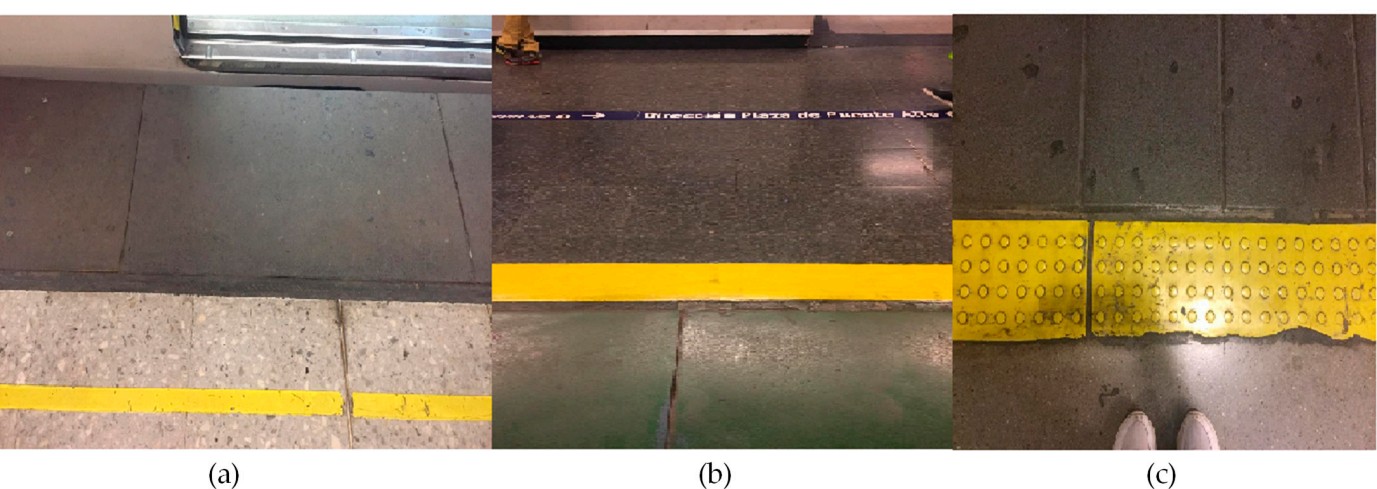

(a) (b) (c)

**Figure 1.** Yellow-safety-line designs in metro stations in Line 1 of the Santiago Metro, Chile: (**a**) 5 cm yellow adhesive tape, (**b**) 10 cm yellow adhesive tape, and (**c**) 24 cm PVC with yellow pods.

In the case of the Valparaiso Metro, the yellow safety line at the edge of the platform is also used as a tactile pavement, which is a resource aimed at facilitating the mobility and orientation of people in the platform–train interface regardless of their physical or cognitive condition, by taking into account perceptive, cognitive, and interaction orientation processes.

A disabled passenger is a person with one or more physical or mental deficiencies, whether due to cognitive or sensory causes, whose participation is prevented or restricted when interacting with various barriers presented in public transport environments [11]. Given this definition, it is possible to conclude that a disabled passenger is defined con-

cerning human capabilities and the characteristics of society. In other words, when there are barriers between human capacities and the elements of society, the full participation of people with these physical or mental limitations in their environment is prevented. According to Alonso [12], to reduce barriers, accessibility should be provided not only to reach a destination (e.g., access to the metro station) but also to use the different pedestrian facilities provided (e.g., platform and turnstiles). When talking about accessibility in the context of disabled passengers, this takes on a much more powerful value, since people with mobility impairments or other types of problem cannot easily move and interact with their environment. As a consequence, large public transport entities such as the Valparaíso Metro [2] and the Santiago Metro [3] comply with wheelchair ramps and elevators when addressing accessibility.

In the case of elderly people, Patil and Raj [13] considered that accessibility is related to the mobility of a passenger, which is understood as a need for physical access. However, accessibility must not only be a physical issue but must also ensure the mobility of people as a whole, and include issues such as inclusion and visible information, among others [14]. Therefore, accessibility does not have a concrete and absolute definition, which can cause confusion, making it difficult to fully understand the concept, often meaning that the total basic needs are not met. Most of the studies related to accessibility are focused on the mobility of people, their ease of walking, and accessibility for all types of people. What is sometimes not taken into account is psychological accessibility. For example, having clear information on how to move to the desired destination, the facilities to understand this information, and measurements of the practices that are already carried out in public transport. Considering the above, the objective of this study was to analyze the effect of yellow-safety-line designs on the behavior of passengers at the platform edge in metro stations. To study the platform–train interface, it was proposed to observe existing stations with a variability of the width and material of the yellow line and to carry out full-scale experiments in a controlled environment, for which passengers with reduced mobility were recruited. The results can be transformed into recommendations to improve the design and safety standards of metro stations, and therefore solve the problem of accessibility.

The rest of the paper is organized as follows: First, existing studies at the platform–train interface are presented. Subsequently, the method used is described based on observation and full-scale experiments. Finally, the results are analyzed, to then propose recommendations and future research are discussed.

## 2. Existing Studies on the Platform–Train Interface

Similarly to other public transport systems worldwide, according to the legal framework in Chile based on the Accessibility Law [11], metro stations need to comply with the following:

- Implement signage that considers different deficiencies that cause disabilities;
- Provide seats and reserved spaces for the use of people with disabilities and reduced mobility in at least one of the cars;
- Include seats for passengers with reduced mobility (10% of the seats), seat signage, preferential seats, and door signage (sound and visual);
- Provide stop request buttons (with a color that contrasts with the surface, with an acoustic and light signal).

These standards are focused on the interior of the vehicles and therefore no clarity is provided on the design that the platforms or other circulation spaces around should have. In the case of cities such as London [15] some manuals address the standards and dimensions for an adequate design of the platform–train interface, specifying variables such as door width, platform length, platform width, and exit routes. However, in the case of Santiago, the Metro company has its standards which are not publicly available. The Santiago Metro in its newest lines (3 and 6), has had an accessibility design, in which loading machines with the Braille system were included and special entrance doors for people with reduced mobility were placed. Inside the train, more seats were added, and preferential



spaces were established to give more space for passengers with reduced mobility. This only applies to these new lines, leaving aside the old lines, which makes this a reason for complaints and confusion by the affected users.

In 2015, a study was carried out on the London Underground to see how the behavior of users varied concerning different designs of the yellow line at the edge of the platform [16]. This study was conducted due to an accident at the platform–train interface, where a woman's coat got caught in the train doors, dragging her along the platform, and causing injuries to her leg and arms. The study was carried out in four stations, and each one had a different type of demarcation. It should be noted that in all stations the same material was used for the yellow line, in addition to the fact that in all of them, the distance between the train and the platform was increased, to have greater security. The study carried out had two main criteria to see if these changes were successful: that no negative effect was seen in delay times, and that acceptance by passengers was maintained or increased. The results of the study showed that delay times were not negatively affected by the variation of the yellow line, which is positive. Likewise, there was no information on any impact on the operations of the stations, which means that the changes did not affect the reliability or congestion of the network. Finally, based on field observations it was shown that fewer people were respecting the yellow line. However, it is worth noting that although compliance was reduced, passengers were generally further away from the edge of the platform than before the test. In addition, the yellow line was least respected when not all the people could get on the train, leaving passengers on the platform. Passengers moved forward to board the train but did not go back behind the yellow line if they could not board.

In the case of Chile, a study carried out by Amestoy [17] defines the standards for universal accessibility and also provides a diagnosis of the Santiago Metro network. This study was carried out in collaboration with SENADIS (National Disability Service), as well as the Metro de Santiago company. SENADIS created the Accessibility Diagnosis Report, better known as the IDA file, which aims to quantify the efforts necessary to generate accessibility conditions that guarantee the right to access and use these facilities by people with disabilities. The IDA file has a specific structure and design; however, it undergoes different variations for each project under evaluation. The variables defined by the IDA tab are the following:

- Accessibility standards in stations: access, ramp, walkways, fixed stairs, escalator, elevator, platforms, intercom, movement corridor, mezzanine, obstacles, tactile guide pavement, ticket office, preferential ticket office, self-service machines, turnstiles, turnstiles for passenger with reduced mobility, platform, tactical band at the edge of the platform, seats on the platform, information screens, sound system, exit doors for passengers with reduced mobility, preferential car, and difference in height or distance between the train and platform;
- Accessibility standards in trains: train doors, preferential cars, preferential seats, preferential spaces, train handrails, train handles, and train intercom.

Other studies have been performed worldwide. For example, accessibility in the Istanbul metro stations was studied [18]. The author used a questionnaire to study two types of people. The first group was made up of five people with visual impairment, while the second group was made up of seven wheelchair-users. Both groups were required to enter the station and then answer some questions. In the case of the platform, the users identified that it is important to use the tactile band for alerting and moving guides. Without this element, users could not take the train or leave the platform.

Similarly, in the Bangkok Metro system, a study was reported through a questionnaire to 600 users, of which 10% were older adults (>60 years old) and 4.5% had some type of mobility or reduced functionality [19]. The authors studied the facilities and connectivity for station users (for example, elevators). In addition, nine indicators were used: psychological (e.g., comfort), temporal (e.g., duration of the trip), affordability (e.g., monthly income), basic needs (e.g., if service responds to needs), connectivity (e.g., mixed use of spaces),

attractive design (e.g., use of services), equity (e.g., access to opportunities), time and activity (e.g., location of activities), and urban environment (e.g., impact caused by the growth of expanded cities). Each user had to rank each accessibility question between 0 (little accessible) and 1 (very accessible).

Prakash et al. [20] examined the technologies used to study the platform–train interface and used a virtual tracking tool to identify when passengers crossed the yellow safety line in metro stations. The use of automatic tracking tools could help to identify if passengers are respecting the yellow safety line. Recently, Aguayo et al. [21] used an algorithm to detect passengers' locations on the platform through laboratory experiments, in which the yellow safety line was an important factor when identifying their behavior. Laboratory experiments have been performed in the last decade to study the behavior of passengers and their accessibility. One of the first experiments was performed by Fernandez et al. [22], in which the vertical and horizontal gaps were studied using a carriage and its adjacent platform. The authors found that behavior and accessibility are affected by the platform height, door width, fare collection method, internal layout of the vehicle, and occupancy of the vehicle. Rudloff et al. [23] reported that the door width is an important element to achieve accessibility at the platform–train interface. Similarly, Fujiyama et al. [24] and Fernandez et al. [25] studied the relationship between the door width and the vertical/horizontal gaps through laboratory experiments. In relation to the level of access, Tyler et al. [26] reported that the use of an elevated platform could improve accessibility at the platform–train interface, however, ramps should have a moderate slope and must not be located in front of a train door. According to Holloway et al. [27] steps also affect behavior and accessibility at the platform train doors. The authors found that passengers presented difficulties boarding and alighting when there are different steps at the train doors. Some authors [8,28] studied the effect of platform-edge doors on the behavior of passengers, in which accessibility is related to the formation of lines of flow, the distance between passengers, and the speed passengers and space used by each passenger. To implement platform-edge doors, level access is needed between the platform and the vehicle. These elements work as barriers to prevent passengers from falling onto the tracks, reducing the number of suicide acts and accidents, due to the doors being closed until the vehicle arrives and before it leaves.

Passengers in waiting areas such as the platform–train interface behave differently from those who are in the circulation zone. For Wu and Ma [29] there are two main types of behavior of passengers who are waiting: queuing or clustering to the side or in front of the doors. Other authors such as Krstanoski [30] considered the whole platform as a waiting area to study the distribution of passengers waiting to board. The author states that the distribution of passengers on the platform depends on various factors: the position of the platform exit at their destination station, the search for the least crowded carriage, how crowded the platform is (e.g., if there is no space to move along the platform passenger will wait near the entrance of the platform), and whether there are markings of the position of doors on the platform. There are also some passengers who are located because of random variables (e.g., meeting with a friend). To reduce the interaction between passengers waiting to board the train, Seriani and Fernandez [31] recommended using a rectangular area or "keep out zone" marked on the platform. In this case, boarding passengers were located outside this rectangular area, preventing them from being an obstacle to those who are alighting. Oliveria et al. [32] studied the distribution of passengers, in which design and technology could motivate passengers not to congregate in front of a particular door when boarding the train. Yang et al. [33] studied the effect of waiting areas on the behavior of passengers boarding and alighting. Seriani et al. [34] reported that the vertical or horizontal gap should be less than 5 cm, especially when considering passengers with reduced mobility. Valdivieso and Seriani [35] studied the space used by a wheelchair-user considering different levels of demand when waiting to board the train. Recently, through laboratory experiments, Seriani et al. [36] studied a vertical handrail for wheelchair users to improve accessibility when boarding and alighting the train. With respect to signs in metro stations, Cheng et al. [37] studied the color of signs in a fire escape. The authors

used eye-tracking and physiological data as indicators during the experiments. Moreover, Seriani and Fujiyama [38] studied the distribution of passengers on the platform by means of real-scale experiments, in which the distance between the platform-edge doors and the yellow safety line was a determinant for the interaction between passengers. Despite the important research done, new experiments are needed to better represent the behavior of various types of passengers and study their accessibility in different spaces. In particular, there is a need to carry out new studies to identify the effect of accessibility elements such as the yellow line on passenger behavior at the platform–train interface, which was the main objective of this paper.

## 3. Experimental Method

In this study, experiments were used based on observations on existing stations. Different metro stations were selected from the Santiago Metro and the Valparaiso Metro to study the platform–train interface.

### 3.1. Definition of Variables

The yellow safety line was studied based on Line 1 of the Santiago Metro, which is composed of 27 stations, from Los Dominicos Station to San Pablo Station, since it is the oldest and busiest in the city [3,4]. For similar reasons, different stations in the Valparaiso Metro were studied, which has only one line from Puerto Station to Limache Station (20 stations) [2]. In both cases, the platform–train interface was selected, which is the space with the highest risk of fatal accidents and high interaction between passengers boarding and alighting [39].

Within the platform–train interface, the observation was completed with videos and photos. The variables observed were classified according to [1]:

- Physical variables: variables related to dimensions of the different circulation elements at the platform–train interface (e.g., width of the yellow safety line);
- Spatial variables: variables that changed the behavior of passengers at the platform–train interface (e.g., material of the yellow safety line);
- Operational variables: variables that affected the operation of the platform–train interface (e.g., number of passengers waiting to board the train).

The variables were measured to study the yellow safety lines in existing stations to then expand the analysis in laboratory experiments, and therefore analyze a range of situations that would be difficult to test in existing scenarios.

### 3.2. Experiments Set-Up

Experiments were carried out by varying the width and material of the yellow safety line in a full-scale model of a train. Based on existing stations, four different scenarios of yellow safety line were defined, in which the platform was 6 square meters (2 m long and 3 m wide). A 1 m length was also set as the safety distance between the yellow safety line and the edge of the platform because in the observed stations, for safety reasons, there is a safety concrete floor of approximately 60 cm before the yellow line (see Figure 2).

The variables that changed in the experiments were the width of the yellow safety line and its material. In total, four scenarios of experiments were performed based on observations in existing stations: (a) width of 5 cm using a yellow adhesive tape; (b) width of 10 cm using a double yellow adhesive tape; (c) width of 24 cm using a PVC material with yellow pods; and (d) width of 40 cm using a carbon and fiberglass reinforced material with yellow pods. These different values of the yellow safety line were based on the field measurements. Figure 3 shows the yellow safety line observed in the Valparaiso Metro which has a similar width (60 cm) and material to that tested in the experiments.

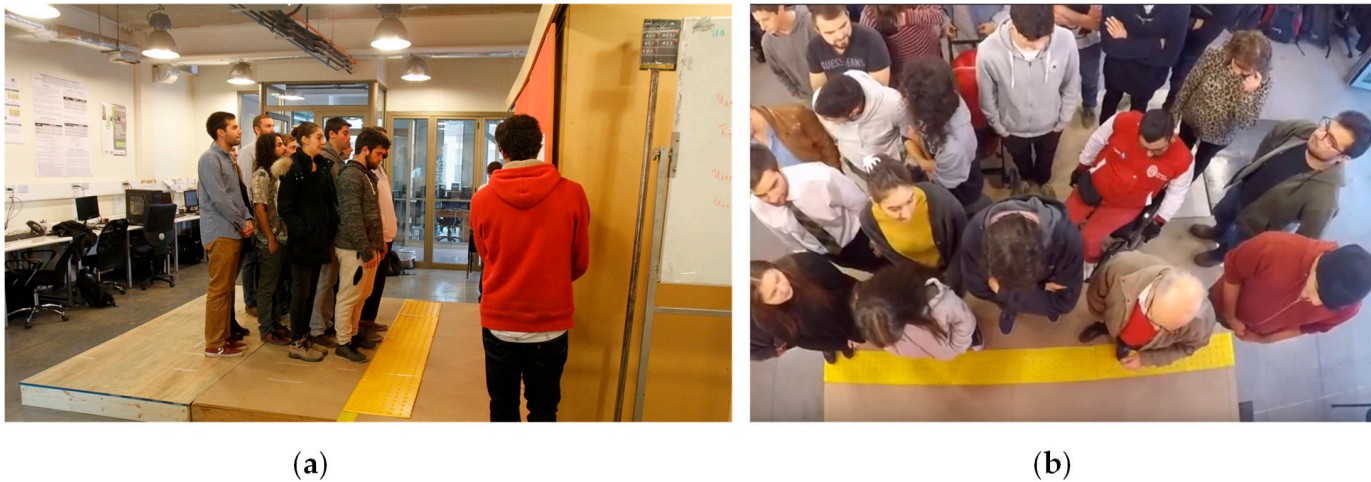

(**a**)          (**b**)

**Figure 2.** Laboratory experiments to represent the boarding and alighting: (**a**) lateral view of the platform when passengers are waiting to board the train and (**b**) top view of the platform when passengers are waiting to board the train.

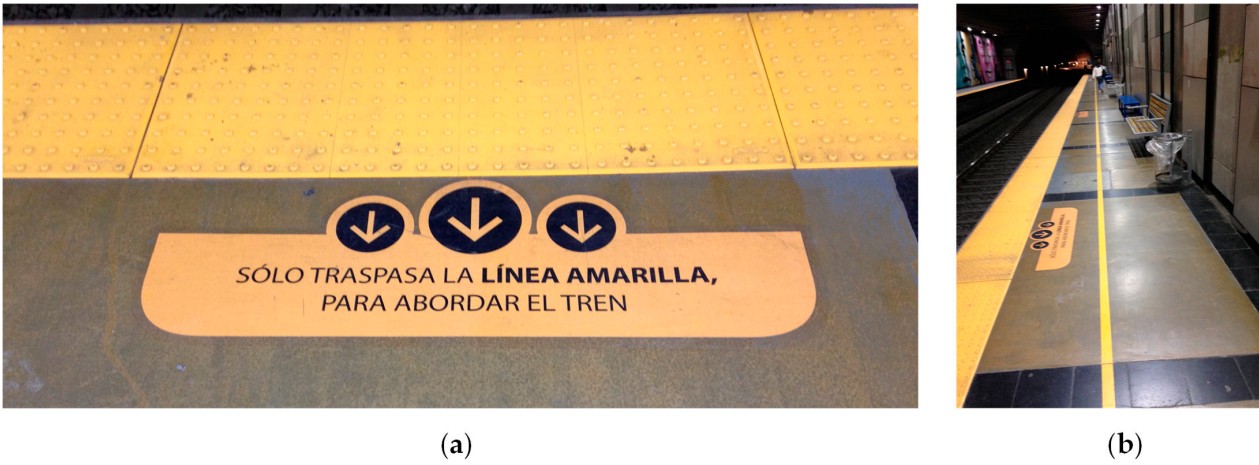

(**a**)          (**b**)

**Figure 3.** Yellow safety line observed in metro stations in the Valparaiso Metro, Chile: (**a**) 60 cm width (carbon and fiberglass reinforced with yellow pods) used with the marking "only cross the yellow safety line when boarding the train" and (**b**) the yellow safety line is used with another yellow line in the middle of the platform to encourage movement on the right-hand side and prevent passengers walking next to the platform edge.

In the experiments, 24 volunteers were recruited to reach a density of 4 passengers per square meter (4 pass/m$^2$), representing a typical situation of the Santiago Metro or the Valparaiso Metro during the rush hour, where these high-density values are reached. From the total number of participants, 3 people had reduced mobility:

- One person used a wheelchair;
- Another person carried a pram;
- One elderly person had walking and hearing problems (hemiparesis).

All the participants arrived at the platform randomly through its entrance, which was previously known to them. Once all the participants got on the platform, the doors of the train were opened so that people could get on and off the train, representing the boarding and alighting process. This was repeated 10 times per scenario since we wanted to study the users' behavior by varying the width of the yellow line in each case. The results were analyzed after the experiment was carried out. In addition, the behavior was recorded by

video cameras previously installed on the side of the platform and located at a height of 2.5 m above the platform, thus having different angles of observation.

### 3.3. Experimental Output

Once the experimentation stage of the experiments was completed, the results obtained were analyzed. First, the videos obtained from the experiment video camera were reviewed, from which it could be seen whether the yellow safety line was respected. To find out which line was most respected in the experiment, an experimental evaluation method was used. This showed the level of compliance that there was on the part of the passengers towards the yellow safety line. Table 1 shows the experimental evaluation method used in the experiments which were adapted from the study carried out by the London Underground [16] to study the platform yellow safety lines.

**Table 1.** Experimental evaluation method to detect when passengers were respecting the yellow safety line considering the direction of travel (DOT), adapted from [16].

| Category | Description | Diagram |
|---|---|---|
| 5 | Feet fully behind the yellow line | |
| 4 | Toes on the yellow line | |
| 3 | Toes over the yellow line, but some part of the foot on the line | |
| 2 | Feet fully over the line/walking down the 'corridor' IN the direction of travel (DOT) | DOT |
| 1 | Walking down the 'corridor' AGAINST the direction of travel | DOT |

Table 1 shows the experimental observation categories, considering each scenario in which the width of the yellow safety line was changed. Table 2 shows the level of compliance using the same 1 to 5 scale. This evaluation method for the level of compliance was based on the position of passengers; however, passengers keep moving so the evaluation process was based on the instant of time when the approach of the train was announced.

In addition, participants were asked to complete a survey to obtain their perception of the safety provided by the different types of yellow safety line, assigning a score to each of them with the following scale: 1. Very unsafe, 2. Unsafe, 3. Indifferent, 4. Safe, 5. Very safe (see Figure 4). Participants were also asked about their perception of comfort when using the different yellow safety lines. The scale used was as follows: 1. Very uncomfortable, 2. Uncomfortable, 3. Indifferent, 4. Comfortable, 5. Very uncomfortable (see Figure 4).

**Table 2.** Level of compliance using the experimental evaluation method to detect when passengers were respecting the yellow safety line adapted from [16].

| Compliance Level | Description | Diagram |
|---|---|---|
| 5 | Full compliance with the yellow line |  |
| 4 | A small level of non-compliance with yellow line (e.g., 2–3 people per car length ignoring the line) |  |
| 3 | Compliance is approximately 50% |  |
| 2 | High level of non-compliance (e.g., more people ignoring the line that obeying it) |  |
| 1 | Complete non-compliance |  |

**Figure 4.** Survey questionnaire given to each of the volunteers who participated in the in each yellow-safety-line scenario.

## 4. Results

This section includes the analysis of the results from the observations made in existing stations and the laboratory experiments. The observations were done in different stations from the metros in Santiago and Valparaiso, Chile. In the case of the experiments, a mock-up represented the boarding and alighting in a controlled environment, in which one variable changed (the width and the material of the yellow safety line) while the rest of the variables remained without variation.

The following sections show the results when applying the method explained in Tables 1 and 2. The number of passengers waiting to board the train was counted according to the behavior when respecting the yellow safety line at the platform edge. The level of compliance was ranked from 1 (yellow safety line is less respected) to 5 (yellow safety line is most respected). This was done considering 10 runs per scenario of the yellow safety line in the experimental facility (Section 4.1) and for three days during peak hour (four trains each day) in the case of existing stations (Section 4.2).

### 4.1. Experiments Considering Different Types of Yellow Safety Line

An experiment was carried out in a controlled environment to represent different yellow-safety-line configurations. In this experiment, people with reduced mobility were included, where one passenger (elderly) suffered from hemiparesis, which is defined as a disease, and is technically a decrease in the movement without reaching paralysis in some limb or side of the body (deafness and other hearing problems). In addition, there was a person in a wheelchair and a person who used a pram (both were 24-year-olds).

Another 21 people were students without disabilities or mobility problems, thus making a total of 24 people who get on and off the train. Of the 24 volunteers, 8 used the metro system five or more times a week, including, in this group, the person in a wheelchair and the person with hemiparesis. The participants generated a density on the platform of around 4 passengers per square meter, which was maintained during the 10 repetitions per scenario.

In order to represent a more realistic situation, each volunteer was given a number before starting the experiment. Five random numbers were chosen before starting each repetition, so that the passengers with the chosen numbers should move in a "hurried" way when getting on the platform and entering the train. These passengers varied for repetition of boarding and alighting.

The method of evaluation adapted from [16] was performed in the experiments for each scenario of the yellow safety line. From the results in Table 3, the yellow safety line that is most respected by the participants of the experiment was the 40 cm line, with a score of 4.7. This was followed by the 10 cm line with a score of 4.2 and the 24 cm line with a similar score of 4.1. Finally, the line least respected by passengers was the 5 cm line, with a score of 3.7.

As expected, these results verified that a greater width of the yellow safety line is more respected by passengers. The one with the least width was the one that was least respected by the passengers, gaining the lowest score of the four scenarios proposed. The lines with thicknesses of 10 and 24 cm had an almost identical score, highlighting that in both cases the score obtained was higher than that of the 5 cm line but lower than that of the 40 cm line. Finally, it can be seen that the 40 cm line was respected in most cases, gaining almost the maximum score of 5.

The results obtained from the surveys (see Figure 5) showed that the widths of 5 and 10 cm are the ones that provide less safety for passengers. It should be noted that both lines were made of the same material (adhesive tape). In addition, the 24 and 40 cm lines had a very high score in terms of safety and did not show much variation in each run. It was thought that the 40 cm line, being wider and made of reinforced carbon fiber and glass material, would make the passenger feel safer than the PVC material (24 cm). However, that was not the case, with the scores being almost the same. In this respect, 14 people surveyed thought that the yellow safety line was a safe method to prevent accidents at

the platform–train interface. Several of these responses alluded to the fact that this line increased the distance between the platform and the train. On the other hand, 9 people thought that the 40 cm line was unsafe as passengers tripped or fell due to its tactile texture (pods). Finally, it should be noted that the person with a wheelchair chose the 5 cm one as the safest because it was the one with the least difficulty for him to pass. He also mentioned that the 40 cm line was difficult to pass due to the tactile texture (pods), as it was difficult for him to move when he was alone.

**Table 3.** Experimental evaluation method of the level of compliance for each scenario of the yellow safety line.

| Scenarios of Yellow Safety Line | Run | Level of Compliance | Average |
|---|---|---|---|
| 5 cm yellow adhesive tape | 1 | 4 | 3.7 |
| | 2 | 3 | |
| | 3 | 4 | |
| | 4 | 4 | |
| | 5 | 3 | |
| | 6 | 4 | |
| | 7 | 3 | |
| | 8 | 4 | |
| | 9 | 3 | |
| | 10 | 5 | |
| 10 cm yellow adhesive tape | 1 | 5 | 4.2 |
| | 2 | 4 | |
| | 3 | 4 | |
| | 4 | 5 | |
| | 5 | 4 | |
| | 6 | 4 | |
| | 7 | 4 | |
| | 8 | 3 | |
| | 9 | 5 | |
| | 10 | 4 | |
| 24 cm PVC with yellow pods | 1 | 5 | 4.1 |
| | 2 | 4 | |
| | 3 | 5 | |
| | 4 | 3 | |
| | 5 | 4 | |
| | 6 | 4 | |
| | 7 | 3 | |
| | 8 | 5 | |
| | 9 | 4 | |
| | 10 | 4 | |
| 40 cm with carbon and fiberglass reinforced material | 1 | 4 | 4.7 |
| | 2 | 5 | |
| | 3 | 5 | |
| | 4 | 4 | |
| | 5 | 5 | |
| | 6 | 5 | |
| | 7 | 4 | |
| | 8 | 5 | |
| | 9 | 5 | |
| | 10 | 5 | |

Likewise, in Figure 5 it can be seen that the width of 5 cm was perceived as the least comfortable setting. Surprisingly, the next-least comfortable was the 40 cm line. In addition, the 24 cm line was evaluated as being the best. It was thought that by having greater safety conditions the user would, in turn, have greater comfort, but this was not the case. Of the total number of volunteers, 12 people referred to the 40 cm line as bad, due to its discomfort. Several opinions mentioned that the tactile texture (with pods) was uncomfortable to walk

on and passengers tripped or fell, especially in the case of women with heels. It should be noted that the person with a wheelchair evaluated the 40 cm line as very uncomfortable, considering that the tactile texture (with pods) was uncomfortable to pass. On the other hand, the elderly person who suffered from hemiparesis described it as very comfortable, since he perceived that "safety offers comfort".

| Participant | Level of Comfort in each Scenario | | | | Level of Safety in each Scenario | | | |
|---|---|---|---|---|---|---|---|---|
| | 5cm | 10cm | 24cm | 40cm | 5cm | 10cm | 24cm | 40cm |
| Ángelo Espinoza | 5 | 5 | 3 | 1 | 5 | 3 | 3 | 3 |
| Boris Petrowitsch | 2 | 2 | 4 | 5 | 1 | 1 | 4 | 5 |
| Inés Maluk | 3 | 3 | 3 | 3 | 2 | 2 | 5 | 3 |
| Claudia Vottero | 3 | 3 | 3 | 3 | 2 | 3 | 4 | 5 |
| Isabelle Fett | 3 | 3 | 2 | 1 | 2 | 4 | 5 | 2 |
| Felipe Mizón | 5 | 5 | 5 | 4 | 2 | 4 | 4 | 5 |
| Domingo Herreros | 3 | 3 | 4 | 5 | 3 | 3 | 3 | 3 |
| Camila Jeldes | 2 | 3 | 4 | 4 | 2 | 2 | 4 | 5 |
| Rodrigo Silva | 4 | 4 | 2 | 1 | 3 | 3 | 4 | 4 |
| José Manuel Vargas | 4 | 5 | 5 | 5 | 3 | 4 | 5 | 5 |
| Matías Kulczewski | 2 | 2 | 4 | 4 | 2 | 3 | 4 | 5 |
| Carolina Suarez | 2 | 4 | 4 | 2 | 1 | 2 | 4 | 5 |
| Andrés Vera | 3 | 3 | 5 | 3 | 3 | 3 | 5 | 4 |
| Andrés Silva | 3 | 3 | 3 | 3 | 5 | 3 | 3 | 2 |
| Gabriel Correa | 1 | 3 | 3 | 5 | 1 | 3 | 3 | 5 |
| Pablo Quilaqueo | 2 | 2 | 5 | 5 | 1 | 3 | 4 | 4 |
| Benjamín Ceroni | 5 | 4 | 4 | 2 | 5 | 5 | 4 | 3 |
| Pablo Vargas | 2 | 2 | 4 | 5 | 1 | 1 | 4 | 5 |
| Lucas Grandón | 5 | 5 | 2 | 2 | 1 | 3 | 4 | 5 |
| Benjamín Salgado | 5 | 5 | 3 | 3 | 3 | 4 | 4 | 3 |
| Manuel Juilo | 4 | 4 | 3 | 2 | 3 | 3 | 3 | 3 |
| Matías Carvajal | 3 | 3 | 5 | 3 | 1 | 1 | 5 | 4 |
| Guillermo Acevedo | 3 | 3 | 3 | 3 | 3 | 3 | 5 | 5 |
| Margarita Forster | 2 | 2 | 4 | 5 | 2 | 3 | 4 | 5 |
| Ricardo Fajardo | 3 | 5 | 3 | 3 | 3 | 5 | 3 | 3 |
| Total | 3.16 | 3.44 | 3.6 | 3.28 | 2.40 | 2.960 | 4.000 | 4.040 |

**Figure 5.** Results from the survey questionnaire to each of the volunteers who participated in the experiments in each yellow-safety-line scenario.

*4.2. Observation of the Yellow Safety Line in Metro Stations*

The variables observed were focused on the yellow safety line, which could affect passengers with reduced mobility such as people in wheelchairs, with a cane, or with other mobility aids, and therefore, have an impact on their accessibility.

In the case of Line 1 in the Santiago Metro, 27 stations were studied, of which all of them are underground stations. The results obtained show that 100% of the stations (27 stations) have elevators to access the platform, in which seats are located and the platform is greater than 3.0 wide. The emergency intercom in all stations is at a height of less than 1.2 m, and the distance between the edge of the platform and the yellow safety line is greater than 0.8 m. However, only 15% of the stations (4 stations) have tactile pavement on the platform floor to alert of changes of direction or guide passengers to move along the platform. In addition, only 33% of the stations (8 stations) have a yellow safety line greater than 24 cm.

In the Valparaiso Metro, 6 stations were studied: Limache, Hospital, Miramar, Recreo, Portales, and Baron. Of these, only Miramar Station and Hospital Station are underground stations. The other 4 stations are overground stations at the street level. These 6 stations are representative of the metro system in Valparaiso. From the observations, all stations achieved accessibility on the platform according to the use of the yellow safety line which was greater than 24 cm wide, and all of them have tactile pavement to guide or alert passengers. However, the distance from the yellow safety line to the platform edge is equal to zero. This is because the yellow safety line is located at the border of the platform edge. In addition, only Recreo, Portales, and Baron stations have a platform width greater than 3.0 m, and none of them have emergency intercommunication at the platform. Miramar Station and Hospital Station, have elevators to access the platform as they are underground stations. Similarly, Portales Station has elevators to access the platform due to the mezzanine which is in an elevated infrastructure. However, the other 3 stations (Limache, Recreo, and Baron) did have elevators to access the platform as they are located at street level.

From the observation in existing stations, the behavior of passengers at the platform–train interface was affected differently at different times (see Figure 6):

- Firstly, passengers access the platform, and it was observed that users, such as older people and young people, walked on the platform looking at their cell phones and not noticing the path, which caused them to step on the yellow line. However, due to its porosity, they corrected their path towards the middle of the platform. On the other hand, an orderly distribution of users was observed throughout the platform, in such a way that it allowed all spaces to be used. In the case of seats, they were generally fully utilized, but in some cases not to full capacity due to social distancing. What was striking was the location of the accessible seats, which were placed at the end of the platform in most cases, leading to the behavior of the elderly using the seats that are not considered accessible, since they do not have a backrest or armrest.
- Secondly, when the train approached the station, users who were closest to the edge of the platform, stuck their heads out to look into the tunnel to see if the train was approaching. Although few passengers do so, it is considered a dangerous act as they could lose their balance and fall onto the tracks.
- Thirdly, users board the train. When the train approached the station, users automatically moved to the edge of the platform, and most of those who were seated stood up respecting the yellow safety line. The passengers who were in the mezzanine were ready to go down the stairs and sped up their steps, running down to get on the train before it closed its doors. This behavior caused interference that prevented a comfortable and safe journey with the users who were leaving the platform.

In each instant of time, it was seen that the yellow safety line was the most important element in achieving accessibility and preventing accidents at the platform–train interface. To study if passengers were respecting the yellow safety line, different observations were made using an evaluation method adapted from [16]. The data was taken over three days (Tuesday, Wednesday, and Thursday) at the Hospital Station during the peak hour from

8:00 a.m. to 9:00 a.m. (in which the interval of trains is 6 min) and off-peak hour from 12:00 to 1:00 p.m. (in which the interval of trains is 15 min). It is worth mentioning that people who use this station should wait behind the yellow line for the users who are inside the vehicle to get off and then go through the yellow line to board the train. That does not occur for all passengers, as a result of wanting to enter the service promptly to reach their destination.

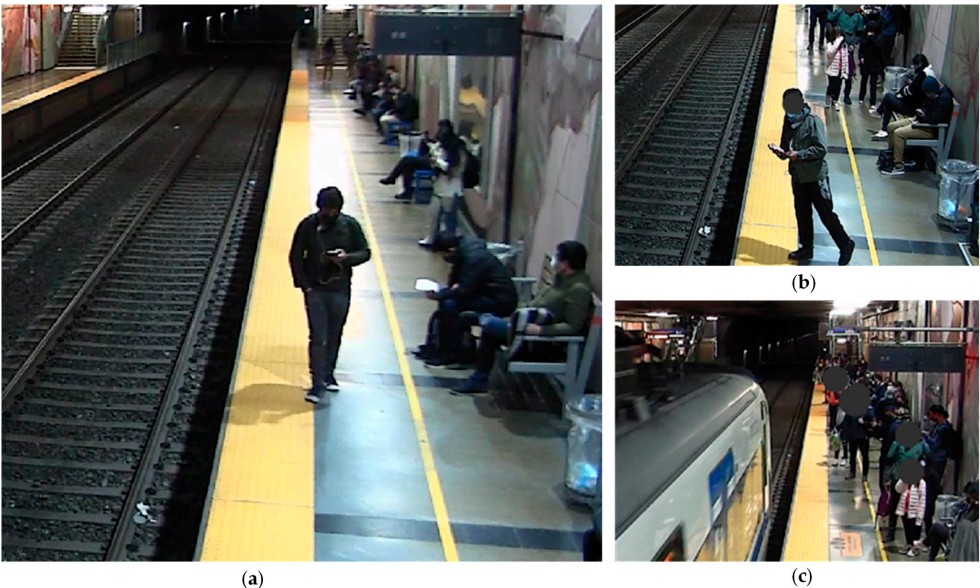

**Figure 6.** Instants of time to study the behavior at the platform train interface in Valparaiso Metro: (**a**) first instant of time in which passengers access the platform; (**b**) second instant of time when passengers walk closer to the yellow safety line; and (**c**) third instant of time when the train approaches and passengers move toward the yellow safety line at the platform edge.

Table 4 shows that on Day 2 there was a higher level of compliance by users who used the service during the off-peak hour. On the day with the highest level of compliance, and an average of 4.25, different types of passengers were registered (older adults, people with mobility issues or difficulties using an elbow cane, university and secondary-education students, and workers). On Day 1, the average was 4.0, however, few passengers with reduced mobility were registered. On Day 3 the average was 3.0, mainly due to the young users who were observed.

**Table 4.** Evaluation method of the level of compliance for the yellow safety line in Hospital Station in the Valparaiso Metro in off-peak hour.

| Day | Train | Level of Compliance | Average |
|---|---|---|---|
| 1 | 1 | 4 | |
| | 2 | 4 | |
| | 3 | 4 | 4.0 |
| | 4 | 4 | |
| 2 | 1 | 4 | |
| | 2 | 4 | |
| | 3 | 5 | 4.25 |
| | 4 | 4 | |
| 3 | 1 | 4 | |
| | 2 | 4 | |
| | 3 | 3 | 3.0 |
| | 4 | 1 | |
| Average of the 3 days | | | 3.75 |

Similarly, Table 5 shows the level of compliance in the case of the peak hour. Here, the highest level of compliance was recorded for Day 3, with a score of 4.2. For Day 2 the score fell to an average of 4.1 and, finally, on Day 1, it averaged 4.0. In this schedule, less diversity was observed concerning the type of passengers, with only students, adults, and mothers with their young children registered.

**Table 5.** Evaluation method of the level of compliance for the yellow safety line in Hospital Station in the Valparaiso Metro in the peak hour.

| Day | Train | Level of Compliance | Average |
| --- | --- | --- | --- |
| 1 | 1 | 4 | 4.0 |
| | 2 | 4 | |
| | 3 | 4 | |
| | 4 | 4 | |
| | 5 | 4 | |
| | 6 | 4 | |
| | 7 | 4 | |
| | 8 | 4 | |
| | 9 | 4 | |
| | 10 | 4 | |
| 2 | 1 | 4 | 4.1 |
| | 2 | 4 | |
| | 3 | 4 | |
| | 4 | 4 | |
| | 5 | 4 | |
| | 6 | 4 | |
| | 7 | 4 | |
| | 8 | 4 | |
| | 9 | 5 | |
| | 10 | 4 | |
| 3 | 1 | 4 | 4.2 |
| | 2 | 4 | |
| | 3 | 4 | |
| | 4 | 4 | |
| | 5 | 5 | |
| | 6 | 4 | |
| | 7 | 4 | |
| | 8 | 4 | |
| | 9 | 4 | |
| | 10 | 5 | |
| Average of the 3 days | | | 4.1 |

## 5. Conclusions

This research sought to analyze the effect of yellow-safety-line designs on the behavior of passengers at the platform edge in metro stations. To this end, the yellow safety line was studied through full-scale experiments based on observations at existing stations in the Santiago Metro and the Valparaiso Metro, Chile.

A method adapted from [16] was applied in the experiments. When analyzing the results obtained, it can be concluded that the yellow line that met requirements both in terms of safety and comfort was the 24 cm line (PVC with yellow pods), which should be considered a standard [11,17] and is present in some stations of Line 1 of the Santiago Metro. In addition, the 5 and 10 cm line did not provide safety to passengers, so it did not provide comfort either (the phenomenon of "safety offers comfort"), except for the person with a wheelchair, since this line did not include a tactile texture. It was also observed that many people believed that the wider the yellow line, the greater the space between the train and the platform. This suggests that a greater width of the yellow line makes users feel a greater distance, making them respect the line more, and at the same time increases the feeling of safety. The 40 cm yellow line did not feel comfortable for many volunteers in

the experiment, which suggests that it is not a good alternative for the metro system. This type of yellow safety line (40 cm) is perceived as unsafe by people in wheelchairs since the tactile pavement (width pods) produced vibrations and passengers may trip or fall when passing over. Even some people without disabilities or reduced mobility feel the same way. When looking at the results, it can be concluded that the 24 cm line is the ideal one to be used.

In the case of the Santiago Metro, it was observed that most of the stations reached the variables defined to achieve accessibility on the platform according to the Accessibility Law [11]. It is important to note that the yellow safety line varied according to different widths and materials. However, in the case of the Valparaiso Metro, the platform of the Hospital Station, the level of compliance depended on the platform congestion. Using the method adapted from [16], the yellow safety line did not reach 100% because users, generally young passengers, when the train approaches, stand on the yellow safety line and later go back and locate themselves in a safe area closer to the platform edge until the door is opened to board the car. Therefore, this highlights that the characteristics of the yellow safety line at the edge of the platform allow people who step on the yellow line to get an alert to prevent them from falling onto the train tracks.

Finally, as future recommendations, it is proposed to study other variables that affect passenger accessibility in metro stations, considering the different spaces and types of passenger that are present in a metro station. Although both the Santiago Metro and the Valparaiso Metro have improved their accessibility [2,3], it is not enough for people with different disabilities or reduced mobility, since, as seen in this study, accessibility depends on both physical and functional variables.

**Author Contributions:** Conceptualization, S.S., A.V., D.L. and T.F.; methodology, S.S. and T.F.; software, P.A., C.B. and V.B.; validation, P.A., C.B. and V.B.; formal analysis, S.S., P.A., C.B. and V.B.; investigation, S.S., A.V., D.L. and T.F.; resources, S.S.; data curation, P.A., C.B. and V.B.; writing—original draft preparation, S.S.; writing—review and editing, P.A., C.B. and V.B.; visualization, P.A., C.B. and V.B.; supervision, S.S., A.V. and D.L.; project administration, S.S.; funding acquisition, S.S. All authors have read and agreed to the published version of the manuscript.

**Funding:** This research was funded by FONDECYT 11200012 and FONDEF id22i10018—both research projects from ANID, Chile.

**Institutional Review Board Statement:** The study was conducted according to the guidelines of the Declaration of Helsinki, and approved by the Institutional Review Board (or Ethics Committee) of Universidad de Los Andes (protocol code CEC202089 approved on 23 October 2020) and Pontificia Universidad Católica de Valparaíso (protocol code BIOEPUCV-H 548-2022 approved on 3 October 2022).

**Informed Consent Statement:** Informed consent was obtained from all subjects involved in the study.

**Data Availability Statement:** Not applicable.

**Acknowledgments:** The author would like to thank the volunteers who represented the boarding and alighting process in the laboratory facility.

**Conflicts of Interest:** The authors declare no conflict of interest.

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
