# Peer review of "Analyzing the Effect of Yellow Safety Line Designs at the Platform Edge in Metro Stations: An Experimental Approach"

_applsci, doi:10.3390/app13084791_

Round 1

Reviewer 1 Report

The paper titled “Analyzing the effect of yellow safety line designs at the platform edge in metro stations: an experimental approach” is reviewed and the authors must revise the paper before it is going to be accepted.

1. The authors must discuss the most relevant literature from the reputed publications in introduction section.

2. Tables 3-5 are difficult to understand.

3. References must be updated with the most recent articles last 5 years.

4. The paper must be minutely checked for language corrections.

Author Response

Review Comments

Reviewer 1

The paper titled “Analyzing the effect of yellow safety line designs at the platform edge in metro stations: an experimental approach” is reviewed and the authors must revise the paper before it is going to be accepted.

Comment 1: The authors must discuss the most relevant literature from the reputed publications in introduction section.

Answer to Comment 1: Thank you for this comment and for reviewing our paper. We added another two papers to expand the literature review:

[37] Chen, N., Zhao, M., Gao, K., Zhao, J. (2020). The physiological experimental study on the effect of different color of safety signs on a virtual subway fire escape—An exploratory case study of zijing mountain subway station. International journal of environmental research and public health, 17(16), 5903.

[38] Seriani, S., Fujiyama, T. (2019). Modelling the distribution of passengers waiting to board the train at metro stations. Journal of Rail Transport Planning & Management, 11.

In addition, we moved references [11] to [14] from section 2 to section 1 to better explain the gap in knowledge presented.

Comment 2: Tables 3-5 are difficult to understand.

Answer to Comment 2: Thank you very much for this comment. We added the following paragraph to better explain Tables 3-5: “The following sections show the results when applying the method explained in Tables 1 and 2. It is counted the number of passengers waiting to board the train according to the behavior when respecting the yellow safety line at the platform edge. The level of compliance is ranked from 1 (yellow safety line is less respected) to 5 (yellow safety line is most respected). This is done considering 10 runs per scenario of the yellow safety line in the experimental facility (section 4.1) and for three days during peak hour (4 trains each day) in the case of existing stations (section 4.2).”.

Comment 3: References must be updated with the most recent articles last 5 years.

Answer to Comment 3: Thank you for this comment. We added recent articles. Please see Answer to Comment 1.

Comment 4: The paper must be minutely checked for language corrections.

Answer to Comment 4: The paper was reviewed and checked in terms of grammar mistakes.

Reviewer 2 Report

Authors analyzed the effect of yellow safety line design at the platform in metro stations in Santiago and Valparaiso, Chile. They analyzed four different yellow safety lines and evaluated level of compliance as well as level of comfort and level of safety of participants. Experimental methods are suitable described. By analyzing result authors conclude that the best yellow line is the 24 cm width.

The used literature is actual and appropriately cited. The images are explained in text and in good quality.
My comments:
The authors monitored the behavior of passengers for 3 days during peak hours and during off-peak hours. But it is not clear which days of the week those were.
In Figure 4, I would consider including the names of the participants rather than just their number. Further, is the number of participant in survey sufficient? Did the authors also evaluate the reliability of the questionnaire?

Overall I think that article can be published after minor revisions.

Author Response

Review Comments

Reviewer 2

Authors analyzed the effect of yellow safety line design at the platform in metro stations in Santiago and Valparaiso, Chile. They analyzed four different yellow safety lines and evaluated level of compliance as well as level of comfort and level of safety of participants. Experimental methods are suitable described. By analyzing result authors conclude that the best yellow line is the 24 cm width.

The used literature is actual and appropriately cited. The images are explained in text and in good quality.

My comments:

Comment 1: The authors monitored the behavior of passengers for 3 days during peak hours and during off-peak hours. But it is not clear which days of the week those were.

Answer to Comment 1: Thank you for this remark. We added the following sentence in section 4.2: “The data is taken over 3 days (Tuesday, Wednesday and Thursday) at the Hospital Station during peak hours from 8:00 a.m. to 9:00 a.m. (in which the interval of trains is 6 minutes) and off-peak hours from 12:00 to 1:00 p.m. (in which the interval of trains is 15 minutes)”.

Comment 2: In Figure 4, I would consider including the names of the participants rather than just their number. Further, is the number of participant in survey sufficient? Did the authors also evaluate the reliability of the questionnaire?

Answer to Comment 2: Thank you for this comment. The names of the participants are included in Figure 4 (first column showed in blue). This study is considered as a pilot study in which we included the details of the participants. From the total number of participants in the experiments, 3 people had reduced mobility:

  • One person who used a wheelchair.
  • Another person who carried a pram.
  • One elderly person who had walking and hearing problems (hemiparesis).

Another 21 people were considered in the experiments who were students without disabilities or mobility problems. However, further studies should be considered to expand this method to existing stations, considering other types of passengers.

Considering the above, we included the following sentence in the conclusions: “Finally, as future recommendations, it is proposed to study other variables that affect passenger accessibility in metro stations, taking into account the different spaces and types of passengers that are present in a metro station.”

Comment 3: Overall I think that article can be published after minor revisions.

Answer to Comment 3: Thank you very much for this comment and for reviewing our paper.

Reviewer 3 Report

The authors have conducted a detailed study of related issues in the field of public safety. This paper has analyzed the effect of yellow safety line designs on the behavior of passengers at the platform edge in metro stations.

From the perspective of academic research, it can be seen that the design of the whole research is reasonable and convincing. Through a large number of tests participated by the volunteers, the experiment produced a lot of valuable data. It can be seen that the argument in this paper is very rigorous and has sufficient experimental support.

Author Response

Review Comments

Reviewer 3

The authors have conducted a detailed study of related issues in the field of public safety. This paper has analyzed the effect of yellow safety line designs on the behavior of passengers at the platform edge in metro stations.

Comment 1: From the perspective of academic research, it can be seen that the design of the whole research is reasonable and convincing. Through a large number of tests participated by the volunteers, the experiment produced a lot of valuable data. It can be seen that the argument in this paper is very rigorous and has sufficient experimental support.

Answer to Comment 1: Thank you very much for this comment and for reviewing our paper.